# Post-COVID-19 Sydrome and Decrease in Health-Related Quality of Life in Kidney Transplant Recipients after SARS-COV-2 Infection—A Cohort Longitudinal Study from the North of Poland

**DOI:** 10.3390/jcm10215205

**Published:** 2021-11-08

**Authors:** Agnieszka Malinowska, Marta Muchlado, Zuzanna Ślizień, Bogdan Biedunkiewicz, Zbigniew Heleniak, Alicja Dębska-Ślizień, Leszek Tylicki

**Affiliations:** Department of Nephrology Transplantology and Internal Medicine, Medical University of Gdańsk, 80-210 Gdańsk, Poland; 34411@gumed.edu.pl (A.M.); marta.muchlado@gumed.edu.pl (M.M.); zuzanna.slizien@gumed.edu.pl (Z.Ś.); bogdan.biedunkiewicz@gumed.edu.pl (B.B.); zbigniew.heleniak@gumed.edu.pl (Z.H.); adeb@gumed.edu.pl (A.D.-Ś.)

**Keywords:** COVID-19, kidney transplant recipient, post-COVID-19 syndrome, long COVID-19 symptoms

## Abstract

Introduction: Patients after SARS-CoV-2 infection frequently face “Post-COVID-19 Syndrome”, defined by symptoms that develop during or after COVID-19, continue for more than 12 weeks, and are not explained by an alternative diagnosis. We aimed to evaluate the presence of post-COVID-19 syndrome and its predictors in kidney transplant recipients (KTR) 6 months after the disease. Materials and Methods: A total of 67 KTR (38 m) with a mean age of 53.6 ± 14 years, 7.3 ± 6.4 years post-transplant were included in the cohort longitudinal study. Thirty-nine (58.2%) of them were hospitalized, but not one required invasive ventilation therapy. They were interviewed 6 months after being infected, with a series of standardized questionnaires: a self-reported symptoms questionnaire, the modified British Medical Research Council (mMRC) dyspnea scale, EQ-5D-5L questionnaire, and EQ-VAS scale. Results: Post-COVID-19 syndrome was diagnosed in 70.1% of KTR and 26.9% of them reported at least three persistent symptoms. The most common symptoms were fatigue (43.3%), hair loss (31.3%), memory impairment (11.9%), muscle aches, and headaches (11.9%). Dyspnea with an mMRC scale grade of at least 1 was reported by 34.3% patients vs. 14.9% before infection; 47.8% stated that they still feel worse than before the disease. Mean EQ-VAS scores were 64.83 vs. 73.34 before infection. The persistent symptoms are more frequent in older patients and those with greater comorbidity. Conclusions: Persistent symptoms of post-COVID-19 syndrome are present in the majority of KTR, which highlights the need for long-term follow-up as well as diagnostic and rehabilitation programs.

## 1. Introduction

There is evidence that a substantial proportion of patients who have been infected with SARS-CoV-2 continue to experience a wide range of symptoms and complications after their acute illness. The most common persisting symptoms include fatigue, muscle and joint pain, shortness of breath, headache, cough, chest pain, altered smell and taste, diarrhea, cognitive impairment, anxiety, and sleep disorders. The complications may be respiratory, cardiovascular, cerebrovascular, neurological, thromboembolic, autoimmune, among others [1,2,3]. Clinicians worldwide have coined the term for the persistent cluster of these symptoms and abnormalities as “post-COVID syndrome”, which may be further sub-categorized into either acute or chronic subtypes, depending on whether symptoms extend beyond 12 weeks following the initial diagnosis [4]. The pathophysiology of these sequelae has been supposed to involve a robust innate immune response with inflammatory cytokine production, cellular damage, and a pro-coagulant state induced by SARS-CoV-2 infection [5]. Data from the literature regarding the prevalence of these complications are inconclusive. In the meta-analysis of 21 studies including 47.910 individuals (hospitalized and not hospitalized), 80% of subjects with a confirmed COVID-19 diagnosis were reported to continue having at least one symptom beyond two weeks following acute infection [6]. The results of another meta-analysis showed that 63.2%, 71.9%, and 45.9% of the sample, including 15.244 hospitalized and 9.011 non-hospitalized patients, exhibited ≥one post-COVID-19 symptom at 30, 60, or ≥90 days after onset/hospitalization [7]. Kidney transplant recipients (KTR) recipients are a high-risk group for severe COVID-19, whose case fatality ratio may reach up to 30%, owing not only to their chronic immunosuppression, but also to older age and frequently associated comorbidities, including hypertension, diabetes mellitus, chronic kidney disease, and cardiovascular complications [8,9]. Contrary to the general population, data on the long-term complications of COVID-19 in this particularly vulnerable group of patients are sparse and are of an exploratory nature [10,11]. Therefore, we aimed to evaluate the presence of post-COVID-19 syndrome and its predictors in KTR 6 months after the disease. We supposed that many of them have post-COVID-19 syndrome, which limits their quality of life and requires further diagnosis and therapy.

## 2. Materials and Methods

### 2.1. Design and Participants

A cohort longitudinal study was performed in all KTR under the care of our institution. All KTR with a diagnosis of COVID-19 confirmed by an RT-PCR test from nasopharyngeal/oropharyngeal swabs until 31 March 2021 were eligible. We excluded the following patients: (a) those who died before the follow-up interview, (b) those for whom follow-up would be difficult owing to a psychotic disorder or dementia, (c) those we were unable to contact. The remaining KTR who consented were included in the study. Their baseline demographic and clinical characteristics, comorbidities and data from the COVID-19 hospital admissions were obtained from the electronic patient records. The Charlson Comorbidity Index (CCI) was calculated according to the formula by summing the assigned weights of all comorbid conditions presented by the patients [12]. Ethics approval for the study was obtained at the Medical University of Gdansk (NKBBN/2014/2021). The study is part of the ‘COVID-19 in Nephrology’ (COViNEPH) project focusing on the nephrological aspects of COVID-19, in particular epidemiology, prevention, disease course, and treatment [3,13,14].

### 2.2. Procedures and Questionaires

Six months after diagnosis, all patients who consented to participate in the study were telephone-interviewed by trained medical students with questionnaires investigating specific persistent or emerging symptoms potentially associated with COVID-19 and the quality of their lives, as previously described [3]. They include a self-reported symptoms questionnaire (SRSQ) according to Huang et al. [15], the modified British Medical Research Council (mMRC) dyspnea scale; the EuroQol consisted of two components: a five-dimension five-level (EQ-5D-5L) questionnaire, and the EuroQol Visual Analogue Scale (EQ-VAS). For the SRSQ (Appendix A), participants were asked to report newly occurring and persistent symptoms, or any symptoms worse than before COVID-19 development at the time of the interview [3]. The mMRC scale is a 5-point scale to characterize the level of dyspnea with physical activity with scores ranging from 0–4, where 0 = I only get breathless with strenuous exercise; 1 = I get short of breath when hurrying on the level or up a slight hill; 2 = I walk slower than people of the same age on the level because of breathlessness, or I have to stop for breath when walking at my own pace on the level; 3 = I stop for breath after walking 100 m or after a few minutes on the level; 4 = I am too breathless to leave the house or I am breathless when dressing [3,16]. The EuroQol is a validated questionnaire which has two components. The first EQ-5D-5L, is a health state classification system with five dimensions: mobility, self-care, usual activities, pain or discomfort, and anxiety or depression, where each can be described by five severity levels ranging from 1—“no problems” to 5—“unable to/extreme problems” [3,17]. The second EQ-VAS is the subjective rate of overall health ranging from 0 to 100 labelled as “the worst health you can imagine” and “the best health you can imagine”, respectively [3,18]. In mMRC and EQ-5D-5L, respondents were asked to describe the severity of problems before COVID-19—retrospectively, and prospectively at the time of completing the questionnaires, 6 months after recovery: “symptoms at this moment”.

### 2.3. Statisitcs

Data were presented as means ± standard deviations for continuous variables, and absolute numbers (percentages) for categorical variables. We report descriptive results, and the sample size was not based on statistical hypothesis testing. The main outcome measures were: (1) the percentage of patients with persistent of COVID-19 symptoms in SRSQ; (2) mMRC score ≥1 in mMRC scale; (3) the percentage of responders reporting no (not any) problem across each of the five EQ-5D-5L dimensions; (4) quality of life in the analog EQ-VAS scale. In the secondary strata analyzes, we evaluated the predictors of post-COVID-19 syndrome. A Chi-square test was used for categorical variables. A *t*-test or Mann–Whitney U/Wilcoxon signed rank tests were used to compare continuous variables where appropriate. *p* < 0.05 (two-tailed) was considered statistically significant. The results were evaluated using the STATISTICA (version 12.0 Stat Soft Inc, Tulsa, Oklahoma, United States software package).

## 3. Results

### 3.1. Patients

Of 1086 adult KTR screened, 189 were confirmed with a COVID-19 diagnosis in their history. Of these, 17 (9.0%) patients died during the acute phase of COVID-19 or during the 6 months after recovery. The screening and patient enrollment process is presented in the flowchart in Figure 1. Finally, 67 participants were enrolled for the questionnaire interview six months after acute COVID-19. The mean (SD) age of the participants (38m, 29f) was 53.14 ± 14.0 years and transplantation vintage was 7.3 ± 6.4 years. Hospitalization due to COVID-19 was necessary in 39 (58.2%) individuals, with a mean duration of hospital stay of 16.5 ± 9.54 days. No one required respiratory therapy. Table 1 shows the detailed characteristics of the study population.

### 3.2. Self-Reported Symptoms (SRSQ)

Six months after SARS-CoV-2 infection, 32 (47.8%) respondents assessed their health as worse compared with their pre-COVID-19 status. A total of 45 (67.2%) patients reported at least one persistent COVID-19-related symptom, whereas 18 (26.9%) reported the persistence of three or more symptoms. The most common reported symptoms were fatigue or muscle weakness (29—43.3%), hair loss (21—31.3%), myalgia or headaches (8—11.9%), and memory disturbances (8—11.9%). Details are presented in Figure 2. There were no significant changes in the weight of the patients (77.39 ± 16.39 vs. 76.39 ± 17.45 kg).

### 3.3. Self-Reported Dyspnea (mMRC)

The presence of dyspnea symptoms before COVID-19 of a grade at least 1 were reported retrospectively by 10 (14.9%) patients. Nobody had significant dyspnea with a score of at least 3. At the time of evaluation, dyspnea symptoms of a grade at least 1 were significantly more frequent and were reported by 23 (34.3%) (*p* = 0.009). Significant dyspnea with a score of at least 3 was reported by 2 (2.9%) individuals. Overall, 18 (26.9%) patients reported increasing dyspnea compared with their pre-COVID-19 status. Details are presented in Figure 3.

### 3.4. Health Related Quality of Life (EuroQoL)

As presented in Figure 4, the decrease in quality of life non-significantly affected all 5 domains of the EQ-5D-5L questionnaire. The “usual activity” and “pain/discomfort” dimensions were the ones most commonly impaired. A total of 33 (49.25%) patients reported a decrease in their EQ-VAS score. The mean EQ-VAS score was 73.34 ± 15.72 before COVID-19, and this significantly deteriorated to 64.83 ± 18.6 (*p* < 0.001).

### 3.5. Predictors of Post-COVID-19 Syndrome

For the purposes of this study, post-COVID syndrome was defined as the presence of at least one persistent symptom in SRSQ and/or an increased severity of dyspnea on the mMRC scale—not attributable to alternative diagnosis. Thus, post-COVID syndrome was found in 47 (70.1%) patients. The results of strata analyses found that persistent symptoms are more frequent in older patients and those with greater comorbidity (Table 2).

## 4. Discussion

To our best knowledge, this is one of the first studies to present the long-term consequences of COVID-19 in KTR. Our data confirm that, unfortunately, serious sequelae of the disease extend beyond the acute phase of COVID-19, similar as in the general population. Six months after recovery, 67.2% KTR still experienced one or more COVID-19-related persistent symptoms. Importantly, we found post-COVID syndrome in patients with a mild to moderate COVID-19 course. None of our patients required invasive ventilation therapy during hospitalization. So far, only two studies have been published on the post-COVID long-term outcomes in KTR. In the prospective cohort study by Basic-Jukic et al., only 11.53% of 104 KTR who survived acute mild to moderate COVID-19 had no clinical symptoms or were free from any laboratory abnormality during the median follow-up of 64 days (range: 50–76 days) after recovery. Prolonged symptom duration and clinical complications were present in 45.2% of patients, whereas 71.2% of individuals had one or more laboratory abnormalities. Six months after acute COVID-19, most of them significantly improved and had no symptoms. On the other hand, many patients required rehospitalizations for severe complications [10]. Unfortunately, they did not have the same time-point for a check-up for every patient, which makes collective analysis and comparison with other studies difficult. Quite recently, Chauhan et al. reported their investigation regarding the long-term consequences of COVID-19 in KTR from India. Even in individuals with mild course of COVID-19, persistent symptoms and deterioration in quality of life were observed up to 6 months of follow-up. Fatigue, alopecia, sleep disturbances, and appetite loss were the most frequently reported symptoms, and anxiety/depression was the worst affected component of quality of life. Importantly, a rapid resolution of persistent symptoms and improving the quality of life was observed in subsequent periodic analyses [19].

Generally, it is difficult to estimate the prevalence, characteristics, and duration of this new condition called post-COVID syndrome, primarily because there is currently no accepted case definition for post-COVID syndrome and consensus on diagnostic procedures [20]. Previous studies used various diagnostic methods (questionnaires, laboratory or imaging tests) and focused on different groups of patients. Most of the early data on post-COVID syndrome emerged from the follow-up of hospitalized individuals with COVID-19 who had a more severe disease course and, consequently, reported a higher prevalence of persistent symptoms. As a result of this, data on the prevalence of post-COVID syndrome in the general population vary greatly, ranging from 75–93% in hospitalized patients [15,21,22], to 10–20% in patients with a mild course of the disease [23,24]. The results of our study (71% of post-COVID syndrome in KTR) are comparable to those obtained in the convalescences from the general population, 6 months after symptoms onset. Using the same standardized questionnaire, Huang et al. showed in their prospective cohort paper that 76% of convalescents reported still at least one persistent symptom. The vast majority of these were patients with a mild disease course, who did not even require oxygen therapy. The percentage of subjects with COVID-19 syndrome among this subgroup was as high as 81%. A large proportion of them had worsened health-related quality of life, and diminution of functional status compared with their pre-COVID-19 status [15]. On the contrary, the presence of at least one persistent symptom at 6 months post-disease was observed in only 8% of KTR in the recent study from India. Importantly, the studied cohort was significantly younger than our patients. Moreover, the percentage of patients with a severe course of COVID-19 was not large and amounted only to 12% [19].

We found here that fatigue was the most common persistent symptom in KTR that was consistent with data from the long-term follow-up study of Huang et al. in the general population and KTR from India [15,19]. Although our cohort reported feeling more breathless compared with the period before infection, it was not the dominant persistent symptom as presented earlier by Basic-Jukic in KTR and other studies from the general population [1,10]. It should be taken into account, however, that the course of COVID-19 in our cohort was mostly mild without significant respiratory involvement. As in other studies, hair loss, sleep difficulties, myalgia, and memory disturbances were some of the most frequently reported persistent symptoms [1,19]. Hair loss was the second most frequently reported complaint, which in KTR may result from the cumulative effects of COVID-19 consequences, and the side effects of immunosuppressants, in particular tacrolimus [25]. It may be a consequence of any acute illness with fever. For COVID-19, acute telogen effluvium as well as androgenetic alopecia are considered [26]. Taking into account the scale of the phenomenon, it requires further research, in particular regarding the pathogenesis hair loss, as well as methods of prevention and treatment. It is noteworthy that few of our patients have reported de novo gastrointestinal symptoms, such as nausea, vomiting or diarrhea—very common persistent symptoms after COVID-19 in the general population. Perhaps the reason for this is the constant presence of these symptoms in a high percentage of KTR as a consequence of the adverse effects of immunosuppressive drugs [27].

It seems interesting to compare the results with the findings of the similar study carried out by our team in the population of maintenance hemodialyzed patients who have recovered from acute COVID-19. Of 79 individuals—39m (49.4%), 40f (50.6%)— with a median age of 70·0 (64·0–76·5) years with HD vintage of 40 (17.5–88) months, 81% reported at least one persistent symptom at six months after discharge. The most common symptoms were fatigue (47.04%), palpitations (30.14%), sleep difficulties (28.77%), nausea (27.4%), and hair loss (24.66%) [3]. A large percentage of residual symptoms reported by patients in the present study were fully reflected in the self-assessment of their health condition (SRSQ), which was described by almost half of the cohort as worse than before the disease. It also resulted in a significant deterioration in health-related quality of life, seen in the EQ-VAS scoring. Decrease in quality of life affected all domains of EQ-5DL, but the highest increase in health problems was reported for the daily activity domain.

In the study of Basic-Jukic et al., complications were more frequent in KTR with decreased glomerular filtration and those with diabetes mellitus [10]. Older age, female gender, hospital admission at symptom onset, initial dyspnea, and comorbidities were found to be significantly associated with an increased risk of post-COVID syndrome in the general population [6,7,28]. Our study was underpowered to perform such analysis reliably, but we observed in strata that older KTR and those with numerous comorbidities reported persistent symptoms more often. Although women and individuals with a more severe course of COVID-19 seem to be at a higher risk of developing complications from the disease, the differences found in the study did not reach statistical significance.

This study has several limitations. First, the study was uncontrolled, which precludes comparison of the frequency of outcomes with KTR who did not suffer from COVID-19. Unfortunately, almost all studies on this issue to date did not use a control group, permitting the inference of counterfactual outcomes. In one of the few controlled cohort studies including 47.780 residents of England, Ayoubkhani at al. showed that individuals discharged from hospital after COVID-19 had increased rates of multiorgan dysfunction (particularly respiratory and cardiometabolic) compared with a matched control group from the general population without COVID-19 in history [29]. Quite recently, other authors reported that 1 year after acute infection, COVID-19 survivors still had lower health status than non-COVID-19 controls matched for age, sex, and comorbidities [30]. Second, a significant proportion of patients refused to take part in our study, possibly introducing a selection bias. Our participants may be different from those who were not included due to various reasons, such as being more motivated to participate because of unresolved symptoms. On the other hand, it cannot be ruled out that the severe and long course of COVID-19 may have discouraged additional contact with medical staff. A relatively small sample size may make it difficult to determine if a particular outcome is a true finding, and in some cases a type II error may occur. To mitigate these weaknesses, it seems warranted to conduct multicenter collaborative research on this topic potentially involving transplant registries. The assessment of the influence of COVID-19 on distant graft function may also be of interest in this regard. Third, because participants were asked to rate their quality of life and severity of dyspnea before COVID-19 six months after their illness, measurement bias cannot be ruled out as well. Fourth, this study may have obtained less accurate information (mainly) because of the nature of telephone follow-up, compared to face-to-face interview or physical examination. On the other hand, it is one of the first studies to analyze the long-term consequences of SARS-COV-2 infection in KTR. Importantly, we used questionnaires that are widely used and well-validated, which makes the results comparable with analyzes carried out quite recently in the general population [30].

In conclusion, our study showed the high frequency of occurrence of post-COVID-19 syndrome in KTR at 6 months after COVID-19. The persistent symptoms including mainly fatigue, hair loss, dyspnea, memory impairment, muscle aches, and headaches were more frequent in older patients and those with greater comorbidity. We demonstrated also a significant deterioration in health-related quality of life and the highest increase in health problems was reported for the “usual activity” and “pain/discomfort” dimensions. Our results highlight the need for a long-term follow-up of convalescences in this population, for diagnostic and rehabilitation programs.

## Figures and Tables

**Figure 1 jcm-10-05205-f001:**
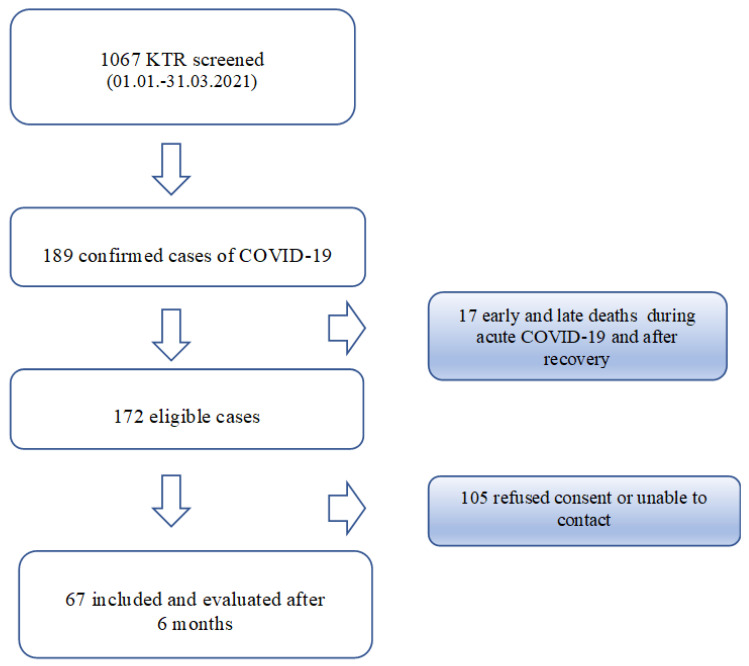
Flow chart of KTR from our institution with SARS-COV-2 infection screened and included in the study.

**Figure 2 jcm-10-05205-f002:**
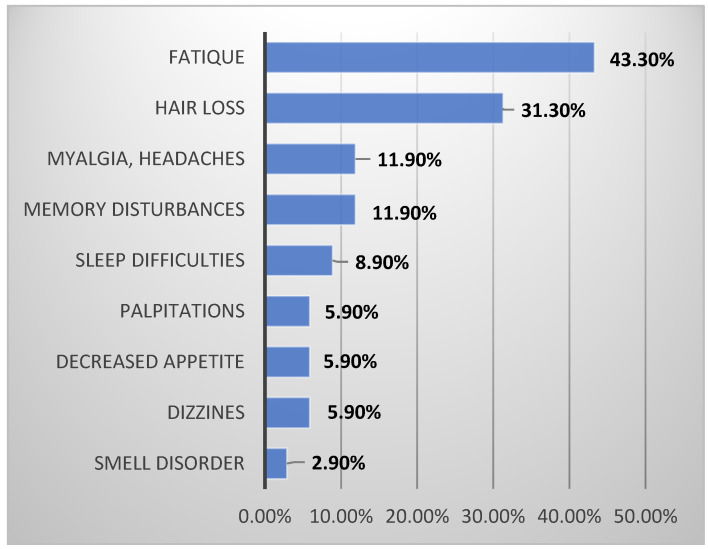
Self-reported symptoms in SRSQ questionnaire.

**Figure 3 jcm-10-05205-f003:**
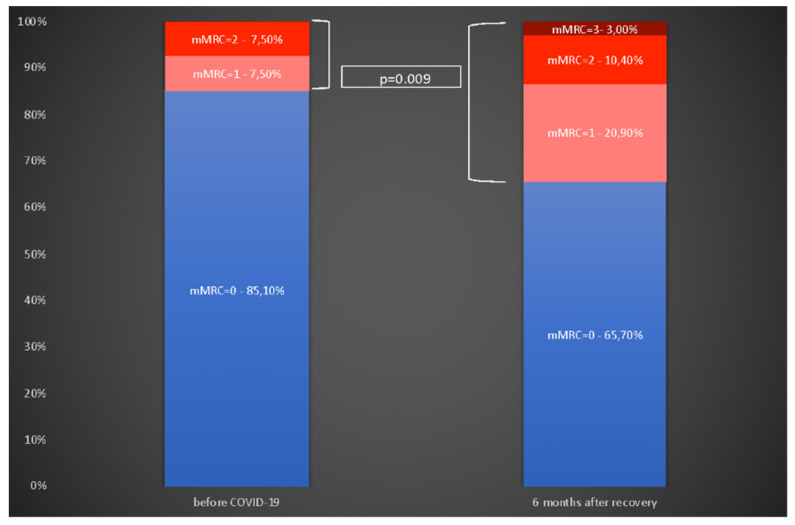
Self-reported dyspnea on exertion (mMRC questionnaire) before COVID-19 and 6 months after recovery. mMRC = 0 (no breathlessness), 1 (breathless when hurrying or walking up a hill), 2 (breathless when walking slower than people of same age or has to stop when walking), 3 (breathlessness stops walking after ∼100 m or a few minutes).

**Figure 4 jcm-10-05205-f004:**
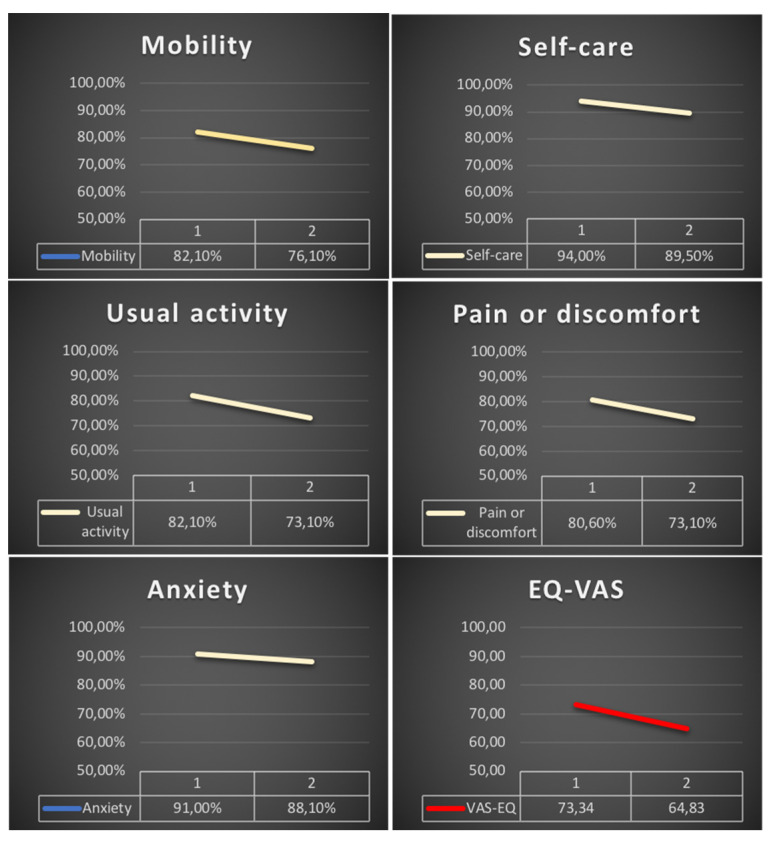
KTR with no problems with quality of life in five dimensions of EQ-5D-5L (yellow lines) and health related quality of life in EQ-VAS questionnaire (red line) before COVID-19 (**1**) and 6 months after recovery (**2**).

**Table 1 jcm-10-05205-t001:** Characteristics of study patients.

*n*	67
Age years mean ± SD	53.14 ± 14.0
Male sex *n* (%)	38 (56.7)
Charlson Comorbidity Index mean ± SD	4.15 ± 2.26
Fragility index mean ± SD	3.25 ± 0.68
Primary nephropathy *n* (%)	
Unknown	13 (19.4)
Glomerulonephritis	23 (34.3)
Diabetes nephropathy	7 (10.4)
Hereditary nephropathies	11 (16.4)
Transplantation vintage years mean ± SD	7.3 ± 6.4
Deceased donor *n* (%)	61 (91.1)
Immunosuppression protocol *n* (%)	
TAC + MMF/MPS + steroids	45 (67.2)
CYS + MMF/MPS + steroids	21 (31.3)
Protocol without steroids	5 (7.5)
Protocol without MMF/MPS	16 (23.9)
SARS-COV-2 infection severity	
COVID-19 asymptomatic	2 (3.0)
COVID-19 symptomatic without hospitalization	28 (41.8)
COVID-19 symptomatic with hospitalization	39 (58.2)
COVID-19 symptomatic with respiratory therapy	0 (0.0)
Total duration of hospitalization due to COVID-19 days mean ± SD	16.5 ± 9.54

Legend: TAC, tacrolimus; MMF/MPS, mycophenolate mofetil/Na; CYS, cyclosporine.

**Table 2 jcm-10-05205-t002:** Strata analyses of predictors for post-COVID syndrome occurrence.

	No Post COVID *n* = 20	Post COVID *n* = 47	*p*-Value
Age years mean ± SD	46.67 ± 16.7	57.02 ± 11.57	*p* = 0.002
Sex woman *n* (%)	6 (30)	22 (46.8)	ns
CCI mean ± SD	3.09 ± 1.3	4.64 ± 2.43	*p* = 0.006
Transplant vintage years mean ± SD	6.21 ± 6.63	7.76 ± 6.3	ns
Serum creatinine mg/dl mean ± SD	1.76 ± 0.92	1.73 ± 1.06	ns
Hospitalization due to COVID-19 *n* (%)	9 (45)	31 (66)	ns

Legend: CCI; Charlson comorbidity index; SD, standard deviation; ns, non significant.

## Data Availability

Detailed data are available on request from corresponding author.

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
