# Peer review of "Post-COVID-19 Sydrome and Decrease in Health-Related Quality of Life in Kidney Transplant Recipients after SARS-COV-2 Infection—A Cohort Longitudinal Study from the North of Poland"

_jcm, 2021, doi:10.3390/jcm10215205_

Round 1

Reviewer 1 Report

See Article review file!

Reviewer 2 Report

This is an important study in that it provides some additional information on the prevalence and scope of Post Covid Symptoms (PCS) in renal transplant recipients 6 months out from their initial infection with the SARS CoV2 virus. Plus how this also has an impact on the quality of life

There are some minor deficits within the manuscript which require attention-

1) The information contained in the second sentence of the 1st paragraph in the Introduction section appears incorrect. I am not sure that there is lots of research implying that there is a 10 fold higher early case fatality. This needs addressing

2) It is clear that there is a recruitment bias in this study looking at how many patients were actually able to be recruited, this needs to be mentioned in the discussion as being one of the limiting factors with respect to interpreting the results. Plus there may also be some selection bias as well(ie patients with symptoms agreed to take part and those without did not)

3) Some of the data elements in the sub section Immunosuppression protocol depicted in Table 1 do not appear to be properly aligned which is confusing to the reader

4) The low case numbers for some of the outcomes mean that although you can detect some trends in the data this is not as yet statistically significant. It would be useful to include in the discussion section what needs to be done to recruit more renal transplant recipients into further research on this topic. ?Multicenter collaborative research and/or involving dialysis/transplant registries. Would this research involve combining qualitative measures with objective measures of ongoing organ impairment due to the sequelae of SARSCoV2 infection?

5) The hair loss is at the upper end of what has been reported in the burgeoning literature on PCS. Noting the rate of hair loss in dialysis patients post COVID is this something which requires further investigation and if so what kind of research would this involve? This is best addressed in the discussion

6) No mention is made of gastrointestinal symptoms which are now known to also be a feature of PCS. Is this because your questionnaire was not set up to detect this subset of symptoms or were they not mentioned by the patients? Mention does need to be made in the manuscript of the gastrointestinal impact of COVID19 and how in a small percentage of patients will have ongoing symptoms (ie this is part of the PCS syndrome)

Reviewer 3 Report

The data presented in the manuscript are interesting and did add a sufficient amount of new information to the data already published in the literature.

 I would recommend accepting this manuscript for publication once the  revisions below have been implemented. 

Some minor grammar mistakes:

  • Line 125 data was evaluated – should be: data were
  • "Post-Covid syndrome" or "post Covid syndrome" – both versions are used in the text – require correction ( eg. lines 199 to 208)

Similar paper was published in September in Transplant Infectious Diseases (DOI: 10.1111/tid.13735) and should be thoroughly discussed in the Discussion section

Author Response

  1. Line 125 data was evaluated – should be: data were

The sentence was corrected (line 120)

  1. "Post-Covid syndrome" or "post Covid syndrome" – both versions are used in the text – require correction ( eg. lines 199 to 208).

The nomenclature was unified. The version – “Post Covid-19” has been left.

  1. Similar paper was published in September in Transplant Infectious Diseases (DOI: 10.1111/tid.13735) and should be thoroughly discussed in the Discussion section.

Thank you very much for this comment. At the time of writing our manuscript, this study was not yet available online. It is quoted and discussed in its current version (line 209, 235).

Round 2

Reviewer 1 Report

Despite the small number and a retrospective manner, I recommend it for publication.

Reviewer 3 Report

All concerns were properly raised.